# Preimplantation Genetic Testing (PGT) to Reduce the Risk for GBA-Related Parkinson’s Disease: Expanding the Applications for Embryo Selection

**DOI:** 10.3390/ijms26030912

**Published:** 2025-01-22

**Authors:** Shachar Zuckerman, Ari Zimran, Jeff Szer, Shoshana Revel-Vilk, Gheona Altarescu

**Affiliations:** 1Medical Genetics Institute, Shaare Zedek Medical Center, The Eisenberg R&D Authority, Faculty of Medicine, Hebrew University of Jerusalem, Jerusalem 9103102, Israel; genetics@szmc.org.il; 2Gaucher Unit, Shaare Zedek Medical Center, The Eisenberg R&D Authority, Faculty of Medicine, Hebrew University of Jerusalem, Jerusalem 9103102, Israel; azimran@gmail.com (A.Z.); svilk@szmc.org.il (S.R.-V.); 3Faculty of Medicine, Hebrew University of Jerusalem, Jerusalem 9112102, Israel; 4Department of Clinical Haematology, Peter MacCallum Cancer Centre, The Royal Melbourne Hospital, Melbourne 3052, Australia; jeff.szer@mh.org.au; 5Department of Medicine, University of Melbourne, Melbourne 3052, Australia

**Keywords:** preimplantation genetic testing, Gaucher disease, GBA variants, Parkinson’s disease

## Abstract

Preimplantation genetic testing (PGT) is practiced worldwide, allowing the prevention of the transmission and expression of various genetic conditions. Socio-ethical considerations of justified applications for PGT are part of an ongoing debate. Pathogenic variants in the glucocerebrosidase (*GBA1*) gene, causing Gaucher disease (GD), have emerged as a risk factor for Parkinson’s disease (PD) in both patients and carriers. Genotype–phenotype correlations exist between different *GBA1* pathogenic variants and the risk to develop PD: mild pathogenic variants increase the risk of developing PD by ~3-fold, while severe pathogenic variants increase this risk by ~15-fold, occurring at a younger age. A woman with GD, a compound heterozygote of N370S (now commonly described as c.1226A>G (N409S)—mild pathogenic variant) and 84insG (severe pathogenic variant), had PGT consulting before planned in vitro-fertilization. Her mother, an 84insG carrier, had early-onset PD. *GBA1* sequencing of her spouse was negative. We discussed the selection for N370S carrier embryos to reduce PD risk. This case report demonstrates the expansion of PGT for late-onset conditions. These novel indications will increase the number of subjects who would be candidates for PGT. The medical and bioethical considerations of these cases should be acknowledged by the professional community and discussed with couples during genetic counseling.

## 1. Introduction

### 1.1. Gaucher Disease

Gaucher disease (GD) is a lysosomal storage disorder, with a carrier frequency of 6% in Ashkenazi Jews [1], compared to an estimated 0.7–0.8% carrier frequency in non-Jews [2,3]. Testing for four variants in the glucocerebrosidase (GBA1) gene [c.1226A>G (N409S) commonly denoted as N370S, 84insG (commonly denoted as 84GG), L444P, and IVS2DS+1G-A (commonly denoted as IVS2+1)] detects at least 96% of carriers in Ashkenazi Jews [2] and about 75% of carriers in non-Jews [3]. GD is caused by deficient glucocerebrosidase enzymatic activity, with the subsequent accumulation of glucosylceramide in various organs [1]. The phenotype in GD is divided into three types: I, II, and III. The commonest phenotype is type I, which predominantly results in visceral manifestations, including thrombocytopenia, hepatosplenomegaly, anemia, and bone disease. In types II and III GD, which are extremely rare in both Ashkenazi Jews, and in the Western Hemisphere in general, there is, in addition, progressive central nervous system disease [4].

The most common GD pathogenic variant in Ashkenazi Jews, N370S, precludes neurological disease, leading to type I GD with an extremely variable phenotype, from symptomatic disease in early childhood to asymptomatic status throughout life [4]. The majority of N370S homozygotes are mildly symptomatic or even asymptomatic [5,6]. A more significant GD expression is usually associated with compound heterozygosity for N370S and a “severe mutation”, e.g., 84insG, L444P, IVS2DS+1G-A, or V394L [4]. The second most frequent Ashkenazi variant, 84insG, is severe but rare (1/400) [2].

In the last few decades, an association between GD and Parkinson’s disease (PD), a neurodegenerative condition which generally becomes clinically manifest in the later years of life, has been identified [7]. Further research has focused on the differential effect of mild versus severe pathogenic variants in *GBA1*, and it has been established that severe variants (e.g., 84insG) increased the risk of developing PD by ~15-fold while mild pathogenic variants (e.g., N370S) increased the risk of developing PD by ~3-fold [8]. The average age at PD onset was also affected by the type of pathogenic variants: carriers of severe pathogenic variants were diagnosed at a mean of 53.1 years compared to carriers of mild pathogenic variants diagnosed at age 58.1 years [8]. Recently, a few very mild variants such as R496H [milder than N370S and E326K (and possibly a polymorphism)] have also been associated with PD, probably at even higher risk than N370S, and accordingly, we suggest that the differential risk of PD related to mild and severe pathogenic variants should be changed to N370S vs. non-N370S [9].

### 1.2. Prenatal Genetic Counseling for GD Patients and Carriers for the Risk of PD in the Fetus

Although the link between *GBA1* pathogenic variants and PD is well established, and a growing number of carriers are being detected through prenatal carrier screening programs, discussion of the risk of the fetus developing PD and preventive options is not part of common practice. A survey of 75 individuals who were screened for autosomal recessive disorders including GD found that 86.7% of respondents believed that patients should be informed about the increased risk of PD prior to GD carrier screening [10].

### 1.3. “Manifesting Heterozygotes”: Risks Associated with Carrying One Mutation of Autosomal Recessive Disorders

In general, for autosomal recessive disorders, heterozygote carriers do not have any phenotype; however, there are several autosomal recessive disorders in which carriers may be at risk for conditions that are different from the homozygote state.

One example is sickle cell disease (SCD)—an autosomal recessive disorder. Heterozygotes of these pathogenic variants, carrying the sickle cell trait (SCT), are at increased risk for various complications common to SCD: hematuria, proteinuria chronic kidney disease, venous thromboembolic disease, and pulmonary embolism [11].

Carriers of pathogenic variants in the CFTR gene, responsible for cystic fibrosis, are also at increased risk of developing a wide range of CF-related conditions across multiple organ systems, including infertility, pancreatitis, sinopulmonary infection, type I diabetes or secondary diabetes, dehydration, electrolyte disorders, constipation, newborn respiratory failure, and short stature [12].

We present a case of risk reduction for late-onset PD by preimplantation genetic testing (PGT).

## 2. Case Report

A couple was referred to the PGT unit from the Gaucher unit in one of our institutes. The woman, of Ashkenazi origin and 38 years old, was diagnosed 20 years previously with GD type 1 [compound heterozygote of N370S (mild variant) and 84insG (severe variant)]. She started treatment with enzyme replacement therapy due to symptoms related to Gaucher disease (anemia and hepatosplenomegaly). Her symptoms improved significantly over a period of less than two years and, at presentation to our clinic, she was asymptomatic. She inherited the N370S variant in the GBA gene from her father (asymptomatic) and the 84insG from her mother who had been diagnosed in her 50s with early-onset PD associated with severe cognitive decline (possibly Lewy Body dementia). No other causes that could explain PD were detected in her mother. Her two twin siblings were wild-type (no mutation) for the familial GBA variants. Her spouse was a healthy 41 year old with no family history of a genetic disorder. No pathogenic variants were detected in the *GBA1* sequence of the spouse. Since the woman was affected with a recessive disorder and her spouse was not a carrier of a pathogenic variant in the same gene, all future offspring will be carriers, having in each pregnancy a 50% chance of inheriting the mild pathogenic variant N370S, and a 50% chance of the severe pathogenic variant, 84insG.

When it became apparent that the couple needed in vitro fertilization (IVF) because of unexplained infertility, the GD unit, in collaboration with the PGT unit, offered a PGT consultation to discuss the possibility of risk reduction for PD by selecting N370S carrier embryos to be transferred. The couple was counseled about the IVF procedure, IVF success rates, the risk of misdiagnosis, and the importance of prenatal diagnosis during pregnancy.

### 2.1. Ovarian Stimulation and Egg Retrieval

IVF treatment was performed using the long down-regulation protocol consisting of 0.1 mg/day of decapeptyl s.c. (Ferring Ltd., Herzliya, Israel) and a daily dose of 300 IU of recombinant FSH (Gonal F; Serono, Israel) and 150 IU of HMG (Menogon; Ferring Ltd.). Follicular tracking was performed using both vaginal and abdominal ultrasonography to enable accurate visualization of the ovaries. An amount of 10,000 IU of HCG (Chorigon; Teva, Petach-Tikva, Israel) was given when at least 3 follicles >18 mm developed. Vaginal ultrasound-guided ovum pick-up was performed 35 h after HCG injection under general anesthesia and oropharyngeal airway insertion, as recommended [13].

### 2.2. Blastomere Biopsy, ICSI, and Embryo Cultures

Cumulus oocyte complexes (COCs) were identified, washed, and transferred to organ culture dishes containing equilibrated culture medium (Medicult, Jyllinge, Denmark) and placed in an incubator with 5% CO_2_. Oocytes were denuded with hyaluronidase (Sigma-Aldrich, St. Louis, MO, USA) 2 h after egg collection and were allowed to recover in the incubator for a further 2 h. Intracytoplasmic sperm injection (ICSI) was then performed on each mature egg. Day-three single-cell blastomere biopsy was performed using the “zona-slitting” technique [14]. Blastomeres were each carefully transferred to separate 0.5 mL tubes containing 5 μL of proteinase K lysis buffer [15]. A sample of culture medium (media blank) from each droplet that contained a biopsied blastomere was analyzed to verify the absence of maternal cellular genetic material or DNA in the culture medium. In addition, a No Template Control (NTC, reaction blank) was used to monitor the absence of external contamination in each PCR reaction.

### 2.3. Molecular Analysis

Genetic testing for N370S/84GG and haplotype analysis was performed with DNA extracted from peripheral blood cells using high salt precipitation [16]. Eight microsatellite polymorphic markers were detected flanking the GBA gene and, of these, four were found to be informative in this family to be used in a multiplex single-cell reaction in order to prevent misdiagnosis due to allele drop out in a single cell [17]. Only samples which were informative for a minimum of 3 polymorphic markers flanking the gene in addition to the two GBA variants were considered for diagnosis. Sterility precautions as previously recommended by the ESHRE PGD consortium were used during all steps [18].

The couple decided to engage in PGT to select for an embryo carrier of the N370S pathogenic variant. The PGT procedure was carried out after a familial haplotype was constructed in the laboratory (Figure 1) [19]. Eight embryos were sampled: four were carriers for 84insG mutation, one was aneuploid, one did not give a conclusive result, and two were N370S carriers and thus transferrable. One embryo was transferred, and pregnancy was achieved. Currently, the couple has had a second baby girl and there is a remaining suitable embryo in storage.

## 3. Discussion

We present the expanded possible indications for PGT by describing a case of PGT with the objective of genetic risk reduction for PD. The application of a PGT procedure in this type of scenario is the subject of ongoing debate in the bioethical community. While performing PGT for lethal and severe genetic childhood-onset diseases is widely accepted, in most countries, ethical consideration starts when discussion begins concerning selecting embryos with non-lethal, treatable, or late-onset conditions, as reflected in the decisions of ethics committees and regulatory guidelines worldwide [20,21,22,23,24,25].

This discussion is a part of wider and more complex issue of acceptable application for prenatal diagnosis (PND), both during pregnancy by chorionic villus sampling (CVS) or amniocentesis and before pregnancy by PGT. Most guidelines regarding PND allow it when there is risk of a severe medical condition. However, the definition of severe condition is debated among medical, philosophical, and sociological groups. When the risk for the fetus is for less severe conditions, i.e., low-penetrant and/or late-onset disease, the option of PGT is usually offered more easily than options of CVS or amniocentesis. Since PGT precludes termination of pregnancy (TOP), some argue that the moral justification for its use should be less strict than for routine PND [26,27,28], considering that the procedure takes place in a very early stage of life (when the embryo is still not considered a fetus), at which point the early embryo has only limited moral value [29]. Adopting this notion allows PGT application for milder medical conditions, including late-onset diseases, cancer predispositions, low-penetrance diseases, and HLA typing. Generally, such conditions are usually not considered justified to be tested for by conservative PND [25], but it is permitted in our country to be tested for by PGT after approval by a local hospital ethics committee. Moreover, when the pregnancy is achieved by IVF, it is argued that this procedure usually includes the selection of “the best embryos” [25]. Parental autonomy regarding decisions over future children is also mentioned as an argument [30,31,32].

Nevertheless, from its beginning, the use of PGT has produced a significant number of ethical arguments regarding a potential slippery slope of social implications which could ultimately lead to eugenics [33,34].

The fundamental question underlying justification for PGT for late-onset conditions is in what ways the quality of life is damaged because of the disease. One of the essential factors in these decisions is the age of onset of symptoms related to the condition. Arguments against PGT for late-onset conditions such as Huntington disease were that the future person at risk for the disease would have a significant number of years of potential normal health, function, and overall well-being before developing any symptoms and, therefore, their life would be worth living [28]. In response to this view, there is a claim that indicating a clear-cut age for developing symptoms, according to which condition is described as serious, is impossible since there is no acceptable time for a genetic condition to occur. Moreover, defining ‘late onset’ is subjective and depends on several factors. Therefore, the purpose of PGT should be to select those fetuses with the best life potential, a definition which includes those who would not suffer from a genetic disease during their lifetime [35].

One of the most discussed uses of PGT from an ethical point of view is hereditary breast and ovarian cancer syndrome (HBOC) in which the risk is induced by carrying BRCA1/2 mutations which increase the risk for breast and ovarian cancer but not before 30 years of age and usually much later. In 2006, the UK Human Fertilization and Embryology Authority (HFEA) approved the use of preimplantation genetic diagnosis (PGDs, as previously named) for this condition [36] and thus initiated an ethical discussion from the viewpoints of both carriers and health professionals [37,38,39].

Recent developments in molecular, cytogenetic, and genomic diagnosis allow the identification of variants that increase, by some degree, the risk for genetic disorders. Carriers of these variants who are applying for embryo selection by PGT raise questions regarding PGT procedures that do not aim to eliminate genetic diagnosis but are indicated to reduce the risk of a specific condition. Furthermore, there are clinics today that are offering a stratification of embryos based on polygenic risk score for diabetes, cardiovascular disorders, and schizophrenia [40]. These scores increase the risk of disease by two- to threefold compared to the general population (still a small risk compared to 25 or 50% risk for a pathogenic variant in recessive and dominant monogenic conditions, respectively).

PGT for copy number variants (CNVs) classified as variants of uncertain significance (VUSs) is another example of the complexity professionals are faced with when asked to perform PGT to select a fetus that might be perfectly healthy throughout their life. In recent years, the demand from couples to select a fetus without VUS findings is growing, while it is apparent that VUS classifications are subject to change [41]. These situations make patient counseling regarding PGT for VUS ethically challenging. Research involving genetic counselors from PGT laboratories explored whether they believed that PGT should be offered for conditions with reduced penetrance or for VUS. All participants reported believing that PGT should be allowable for conditions with reduced penetrance and VUS, while the main ethical consideration when deciding to accept or reject a PGT request was patient autonomy, with a focus on the patient understanding risks of the testing. Nevertheless, genetic counselors declared that having guidelines from a professional organization would benefit their practice [42].

In a recent review aimed to comprehensively report on various aspects of couples’ experiences of PGT, there was a consensus that PGT should be applied to lethal or severe childhood diseases but there was less agreement on PGT for adult-onset or variable expression conditions [43]. A more permissive approach was reflected in a survey of Israeli PGT users who were homogeneous in their attitudes that late-onset conditions are a justified application of PGT [35].

The case we presented reflects a new era of PGT. Polygenic scores for multifactorial disorders are now available and are offered in the US for risk reduction PGT for diseases like diabetes, schizophrenia, and hypertension [40]. Lately, ‘The Clinical Genome Resource Low Penetrance/Risk Allele Working Group’ of the American College of Medical Genetics and Genomics recognized risk alleles and low-penetrance variants that require special considerations regarding their clinical classification and reporting, in the prenatal setting. The common variant N370S in *GBA1* was included in their ‘Risk Allele Reporting Scenario Consensus’ [44].

Applying the technology for risk reduction of medical conditions and not for eliminating disease will broaden the spectrum of applications and the number of potential and actual users. The main challenge in this regard is the genetic counseling given before a reproductive decision is made by potential users of PGT, regarding the risk reduction of the specific condition. The genetic counseling must obtain an understanding of both potential risks of the procedure and the residual risk of the condition tested for the fetus. Additionally, guidelines for PGT usage for preventing genetic traits would help professionals present comprehensive ethical decisions on which PGT applications are both justified and moral. As for using PGT for risk reduction of GBA-related PD, we recommend discussing this with patients with GD or carriers (regardless of the partner) who are referred to IVF for infertility considerations in order to achieve fully informed personal decisions regarding risk assessments and moral considerations. In similar cases where IVF is not required, thorough counseling on the advantages and disadvantages of a relatively invasive procedure and the understanding of the emotional distress of the couple is offered in our clinic.

To our knowledge, there are no reports on PGT risk reduction for PD in GBA carriers of pathogenic variants. A consensus opinion of Gaucher disease experts would be of value regarding prenatal decisions in carriers of GBA variants.

## 4. Conclusions

In conclusion, we present an example that illustrates a potential use of PGT for late-onset conditions. The presence of risk factors for genetic disorders could increase the number of subjects requesting PGT. The medical and bioethical considerations of these cases should be acknowledged by the professional community and discussed with parents during genetic counseling.

## Figures and Tables

**Figure 1 ijms-26-00912-f001:**
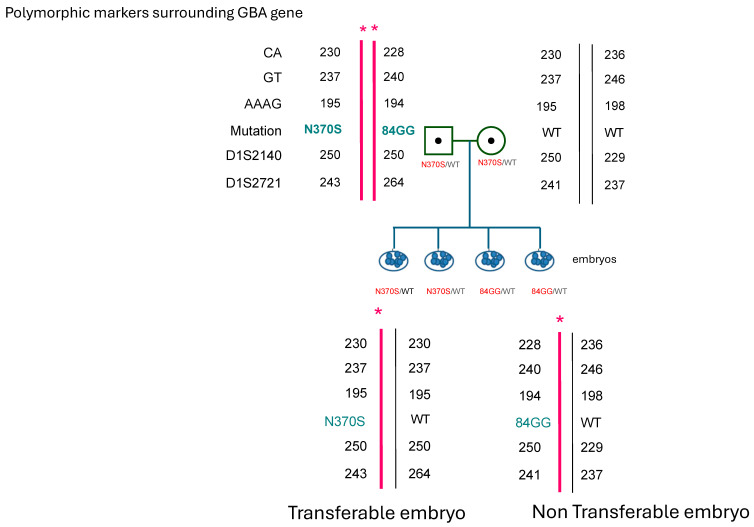
Schematic representation of affected N370S/84GGins in the *GBA* gene of the female partner. Single tandem polymorphic markers used in the diagnosis of the embryos in conjunction with the genetic variant of N370S and example of one affected and one wild-type embryo. * = affected allele.

## Data Availability

The data presented in this study are available on request from the corresponding author.

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
