# Peer review of "Preimplantation Genetic Testing (PGT) to Reduce the Risk for GBA-Related Parkinson’s Disease: Expanding the Applications for Embryo Selection"

_ijms, 2025, doi:10.3390/ijms26030912_

Round 1

Reviewer 1 Report

Comments and Suggestions for Authors

The authors state they are presenting a case study for in vitro selection of preimplantation genetic testing (PGT) to reduce the risk for GBA-related Parkinson disease.  However, only a small fraction of the paper was dedicated as a case report.  Most of the paper reads more like a literature review for PGT testing for late onset diseases. 

As is, this paper does not presently meet the criteria for Case Report. As is, this leans more towards an incomplete literature review on  the ethical concerns of late onset disease embryo selection. For a case report, there are insufficient details on patient details (including demographics, medical history, symptoms, and physical examination findings), diagnosis, treatment plan, outcome and follow-up information, discussion with implications for clinical practice, and a conclusion summarizing the key points of the case.

The paper needs reframed to be a Case Report to: 1) focus on the particulars of this particular carrier’s disease severity, medical and emotional reasoning for undergoing pre-genetic testing for GBA-related Parkinson’s Disease; 2) present the existing protocols in place at the local institution/country of origin regarding any potential ethical restrictions on genetic testing for non-fatal disease; 3) present information on any additional steps taken that mitigated risk of physical or ethical risk; 4) summarize how what was learned/experienced in this case can be generalized to others.

The paper would be greatly improved with an overview decision chart presenting the steps for PGT in light of ethical concerns, risk to mother, embryo, etc.

The first couple of paragraphs of the Introduction seem disconnected from the rest of the work, including odd formatting errors. The authors need to either remove or revise this portion of the Introduction.

The discussion would benefit from better elaborating on differences in ethical and legal protocols for this procedure for the given condition worldwide.

The paper has significant typos and inappropriate/poor English throughout.  Many sentences are quite long and without appropriate punctuation.  Please review and proofread thoroughly.

Comments on the Quality of English Language

The paper has significant typos and inappropriate/poor English throughout.  Many sentences are quite long and without appropriate punctuation.  Please review and proofread thoroughly.

Author Response

The presentation of the case report was substantially changed by adding a significant number of details as suggested.

Regarding the existing protocol for ethical restrictions on genetic testing, this was addressed  in  the Discussion section  lines 219-226 with references 20-25, and an additional  sentence regarding local guidelines was added ( lines242-243).

Concerning steps to address physical or ethical risks, since the couple required IVF regardless, no additional physical risk was introduced. The ethical risk of PGT for late onset disease was thoroughly discussed widely in the introduction.

The generalization of this case was examined in the discussion section, and we added conclusions emphasizing this point.

Regarding the overview of decision- making processes in light of ethical concerns and the risk to the mother and embryo , these points were presented in the introduction and discussion section. Additionally, we included a discussion of  similar diseases in  couples that do not require IVF (lines328-331)

Thank you for drawing our attention to the structure of the introduction.  We have moved the initial paragraphs to the end of the introduction.

Concerning the ethical protocols for this given condtion world wide we added the sentence in lines 332-334.

Regarding language style,  the third author,  J. Szer,  is a native English speaker, and has thoroughly revised the grammar and formatting  of the paper.

Reviewer 2 Report

Comments and Suggestions for Authors

Dear authors, the issue is of great interest.

The introduction is well structured. The study design is described in correctly.

In discussions in the following sentence: “Since PGT precludes termination of pregnancy (TOP), some argue that the moral justification for its use should be less strict than for routine PND [20-22 26-28], considering that the procedure takes place at a very early stage of life” in the pdf version 20-22 looks deleted, please confirm.

A problem I detected is the word concordance in the iThenticate report  which is of 36%. I don t want to associate it with plagiarism.

You also need a Conclusion.

Author Response

References 20-22 were indeed deleted and now corrected in the new uploaded version of the  paper.

We strongly affirm that the paper is original and does not contain any copied material from another scientific  sources or websites.

As suggested, we added a conclusion to the discussion.

Round 2

Reviewer 1 Report

Comments and Suggestions for Authors

The authors have made the major technical changes, which has greatly improved the content of the paper.

Major typos were corrected. However, the grammatical structure of the sentences remained very problematic throughout. A sentence or two here or there is fine. However, the issue is persistent, which takes away from the content of the paper. The major issue is the inappropriate use of compound-complex sentences by punctuating with multiple commas instead of semicolons. Here is one example of many from this paper.

Authors text:

"In general, for autosomal recessive disorders, heterozygote carriers do not have any phenotype, however, there are several autosomal recessive disorders in which carriers may be at risk for conditions that are different from the homozygote state."

Reviewer Suggestion - 

There are two options for a sentence like this.  If you want to keep the two sentences connected, simply use the semicolon per option 1.  Alternatively, you can just separate into two distinct sentences.

Option 1 - Use a Semicolon

In general, for autosomal recessive disorders, heterozygote carriers do not have any phenotype; however, there are several autosomal recessive disorders in which carriers may be at risk for conditions that are different from the homozygote state.

Option 2 - Separate sentences that each have their own subject and verb by punctuating with a period.

In general, for autosomal recessive disorders, heterozygote carriers do not have any phenotype. However, there are several autosomal recessive disorders in which carriers may be at risk for conditions that are different from the homozygote state.

Comments on the Quality of English Language

Major typos were corrected. However, the grammatical structure of the sentences remained very problematic throughout. A grammatically incorrect sentence or two is fine. However, the issue is persistent throughout, which takes away from the content of the paper. The major issue is the inappropriate use of compound-complex sentences by punctuating with multiple commas instead of semicolons. Here is one example of many from this paper.

Authors text:

"In general, for autosomal recessive disorders, heterozygote carriers do not have any phenotype, however, there are several autosomal recessive disorders in which carriers may be at risk for conditions that are different from the homozygote state."

Reviewer Suggestion - 

There are two options for a sentence like this.  If you want to keep the two sentences connected, simply use the semicolon per option 1.  Alternatively, you can just separate into two distinct sentences. Note 'sentence' here is defined as having both a subject and a verb.

Option 1 - Use a Semicolon

In general, for autosomal recessive disorders, heterozygote carriers do not have any phenotype; however, there are several autosomal recessive disorders in which carriers may be at risk for conditions that are different from the homozygote state.

Option 2 - Separate sentences that each have their own subject and verb by punctuating with a period.

In general, for autosomal recessive disorders, heterozygote carriers do not have any phenotype. However, there are several autosomal recessive disorders in which carriers may be at risk for conditions that are different from the homozygote state.